# Dual PPRαϒ Agonists for the Management of Dyslipidemia: A Systematic Review and Meta-Analysis of Randomized Clinical Trials

**DOI:** 10.3390/jcm12175674

**Published:** 2023-08-31

**Authors:** Antonio da Silva Menezes Junior, Vinícius Martins Rodrigues Oliveira, Izadora Caiado Oliveira, André Maroccolo de Sousa, Ana Júlia Prego Santana, Davi Peixoto Craveiro Carvalho, Ricardo Figueiredo Paro Piai, Fernando Henrique Matos, Arthur Marot de Paiva, Gabriel Baêta Branquinho Reis

**Affiliations:** 1Faculty of Medicine, Federal University of Goiás, Goiânia 74605020, Brazil; vinicius.martins@discente.ufg.br (V.M.R.O.); izadora.caiado@discente.ufg.br (I.C.O.); andremaroccolo@discente.ufg.br (A.M.d.S.); ana.santana@discente.ufg.br (A.J.P.S.); davi.peixoto@discente.ufg.br (D.P.C.C.); ricardopiai@discente.ufg.br (R.F.P.P.); fernando.matos@discente.ufg.br (F.H.M.); amarotdepaiva@gmail.com (A.M.d.P.); gabrielbaeta@discente.ufg.br (G.B.B.R.); 2School of Medical and Life Sciences, Pontifical Catholic University of Goiás, Goiânia 74605050, Brazil

**Keywords:** dual PPRαϒ agonist, dyslipidemia, saroglitazar

## Abstract

Saroglitazar is a novel medication for dyslipidemia, but its specific effects remain unclear. Therefore, we performed a systematic review and meta-analysis to assess the efficacy and safety of saroglitazar for managing dyslipidemia. The PubMed, Scopus, and EMBASE databases were systematically searched for randomized controlled trials (RCTs) comparing 2 and 4 mg of saroglitazar with placebos for treating dyslipidemia. A random-effects model calculated the pooled mean differences for continuous outcomes with 95% confidence intervals. The study included seven RCTs involving 1975 patients. Overall, 340 (31.0%) and 513 (46.8%) participants received 2 and 4 mg of saroglitazar, respectively; 242 (22.11%) received the placebo. The mean ages ranged from 40.2 to 62.6 years, and 436 (39.8%) were women. Compared to the control group, 4 mg of saroglitazar significantly decreased the triglyceride and low-density lipoprotein (LDL) cholesterol levels but did not affect the high-density lipoprotein cholesterol level. Furthermore, the alanine aminotransferase level significantly decreased, the creatine level significantly increased, and body weight did not differ between the groups. Finally, 4 mg of saroglitazar, compared to 2 mg, significantly lowered the triglyceride level. Saroglitazar (4 mg) may be an effective treatment, but safety concerns remain.

## 1. Introduction

Diabetic dyslipidemia is characterized by increased triglyceride levels. Specifically, atherogenic dyslipidemia is characterized by a high level of small, dense low-density lipoprotein cholesterol (LDL-C) and a low level of high-density lipoprotein cholesterol (HDL-C) [1]. Together, these are commonly called the “triad of increased triglycerides.” Dyslipidemia is a frequent feature of obesity, metabolic syndrome, insulin resistance, and type 2 diabetes mellitus [2]. It is also a substantial risk factor for myocardial infarction and cardiovascular disease. In addition, small, dense LDL-C particles speed up atherosclerosis progression, leading to an increased risk of cardiovascular disease-related mortality and morbidity [3].

Effective and well-studied treatments for dyslipidemia exist, such as statins and lifestyle modifications [4,5]. Statins, fibrates, and omega-3 fatty acids can manage dyslipidemia. However, although statins reduce cardiovascular events and decrease mortality, a considerable residual cardiovascular risk persists [6]. Moreover, the recent PROMINENT study failed to show pemafibrate reducing cardiovascular outcomes, notwithstanding that the lipid profile was significantly improved [7].

PPARα/ϒ is primarily expressed in the liver, lowering lipotoxicity and circulating atherogenic lipid levels [8]. Saroglitazar, (S)-a-ethoxy-4-(2-methyl-5-(4-methylthio) phenyl)] [(S)-a-ethoxy-4-(2-methyl-5-(4-methylthio) phenyl]-1H-pyrrol-1-yl]-magnesium salt of ethoxy)-benzenepropanoic acid] is a new PPAR/agonist that was produced in India by Zydus Cadila and is marketed under the name Lipaglycn. It was given the green light for use in the treatment of diabetic dyslipidemia and hypertriglyceridemia by the Drug Controller General of India (DCGI) [8]. Saroglitazar also reduces hemoglobin A1c (HbA1c) levels, in part, by lowering lipotoxicity and exerting a modest amount of PPAR-agonistic action in the body. A previous study demonstrated that adding saroglitazar to metformin treatment resulted in a greater decrease in total cholesterol and HbA1c levels compared to that with fenofibrate [9]. However, saroglitazar is controversial because of conflicting findings regarding its safety and efficacy [10]. Krishnappa et al. found no abnormal findings in all treatment groups for serum creatinine, hematocrit, respiratory rate, or body temperature [11]. However, Gawrieh et al. reported a significant increase in serum creatinine levels [12].

Consequently, saroglitazar has only been approved for use in India; other institutions, such as the United States Food and Drug Administration, have yet to approve the medication [10]. Therefore, we performed a systematic review and meta-analysis to investigate the effectiveness and safety of saroglitazar in patients at risk of dyslipidemia and type II diabetes. 

## 2. Materials and Methods

The current systematic literature review was prospectively registered on 25 May 2023 with the International Prospective Register of Systematic Reviews (PROSPERO registration number: CRD42023426614). Results are reported following the Preferred Reporting Items for Systematic Reviews and Meta-Analyses (PRISMA) guidelines [13].

### 2.1. Study Eligibility Criteria

Our primary study question was: “What is the safety and effectiveness of saroglitazar for treating dyslipidemia?” Our investigation was conducted based on the following population, intervention, comparison, outcome, and time (study design) (PICOT) categories: Population (i.e., P): adult patients with diabetic hypercholesterolemia or dyslipidemia; Intervention (i.e., I): saroglitazar at 2 and 4 mg doses; Comparison (i.e., C): placebo; Outcomes (i.e., O): the effectiveness and safety of saroglitazar; and Time (i.e., T): randomized clinical trials (RCTs) lasting more than four months. Efficacy assessments comprised the effects of saroglitazar on total cholesterol, LDL-C, triglycerides, HDL-C, non-HDL-C, and fasting plasma glucose (FPG) levels. Safety assessments included serum creatinine, alanine transaminase (ALT), aspartate transaminase levels, and body weight measurements. The systematic review includes all studies that met the above-mentioned PICOT criteria. Reviews, observational and descriptive research, editorials, comments, and conference proceedings were excluded.

### 2.2. Data Sources and Literature Searches

The PubMed, Scopus, and Embase databases were comprehensively searched from their inception through May 2023; the first search was performed in April 2023. All Medical Subject Heading phrases and keywords linked to “saroglitazar,” “hypercholesterolemia,” and “diabetes” were used in the search [(‘saroglitazar’/exp OR saroglitazar) AND (‘diabetes’/exp OR ‘diabetes’ OR ‘hypertriglyceridemia’ OR‘metabolic syndrome’ OR ‘dyslipidemia’/exp OR ‘dyslipidemia’] to collect previously published studies. The reference lists of the included studies were screened, and a snowball search was performed. The Appendix A present a detailed search strategy for each database.

### 2.3. Study Selection and Data Extraction

The titles and abstracts of the retrieved papers were evaluated, followed by the full text, using the inclusion and exclusion criteria. Extremely irrelevant studies were removed during the title and abstract search, completed by two independent reviewers (IC and VM), followed by the full-text screening. A well-defined data extraction sheet that included information regarding the study’s characteristics, participants, interventions, comparator, and results was used. Two separate researchers (IC and VM) performed the study selection and data extraction, and differences were handled by consensus or discussion with a third reviewer (AM).

### 2.4. Risk of Bias and Quality Assessments

The Cochrane Risk of Bias Assessment Tool [14] was used to assess the methodological quality of the included studies. The studies were divided into low, high, and unclear risk of bias in each domain: randomization, allocation concealment, patient blinding, blinding of outcome measurements, incomplete outcome data, and selective reporting.

The quality of the included double-arm trials was evaluated by the National Institutes of Health Checklist [15], which consists of 12 items identifying the methodological features based on the existing study design and study-reporting guidelines. Each item carries one point, and scores of 0–4, 4–8, and 9–12 were considered poor, fair, and good-quality studies, respectively. Two independent reviewers (IC and VM) evaluated the risk of bias and quality of the included studies and resolved disagreements through consensus or a discussion with a third reviewer.

### 2.5. Data Synthesis

The analyses were performed using RevMan software (Review Manager [computer program], Version 5.4.1, The Cochrane Collaboration, 2004) using a meta-package. Changes in continuous outcomes were calculated for every included study arm by subtracting the value at baseline from the value after the intervention. All of the estimates of effectiveness were presented in the form of mean differences (MDs) or absolute weighted mean changes. A safety estimate, ALT, was presented in the form of standard mean differences (SMD) and 95% confidence intervals (CIs) from the baseline. Based on the Cochrane Handbook, standard deviations (SDs) were calculated from the standard error or 95% CI for a systematic review of interventions [16]. The Higgins I^2^ statistics and Cochran’s Q test were used to assess the potential statistical heterogeneity among trials. The meta-analysis was conducted using a fixed-effect model (using inverse variance) or a random-effect model (DerSimonian-Laird method) based on low heterogeneity (50%). If the low (<50%) and high (>50%) heterogeneity criteria resulted in a low number of studies (<10), publication bias assessments were performed using funnel plots (Appendix A).

## 3. Results

### 3.1. Study Selection and Baseline Characteristics

The electronic database search yielded 267 citations; 196 citations were screened after removing 71 duplicates. Next, 160 studies were excluded during first-pass screening after reviewing the titles and abstracts. The full texts of 36 citations were downloaded for the second-pass screening. Finally, eight articles met our inclusion criteria. Figure 1 presents the PRISMA flowchart for the study selection.

Table 1 summarizes the characteristics of the included studies. Of the 8 included studies, 7 were RCTs, and 1 was a pooled analysis of RCTs. They included 1975 patients, of whom 546 and 731 received 2 and 4 mg of saroglitazar, respectively. Furthermore, 389 and 40 patients received 30 and 45 mg of pioglitazone, respectively. Finally, 18 patients received 160 mg of fenofibrate, and 251 received a placebo. 

Furthermore, five studies [12,17,18,19,20] analyzed the efficacy and safety endpoints between patients receiving saroglitazar and placebo. In 3 studies [9,11,21], the control patients received anti-lipid medications (i.e., 10 mg of atorvastatin, 160 mg of fenofibrate, 30 mg of pioglitazone, and 45 mg of pioglitazone, respectively).

**Table 1 jcm-12-05674-t001:** Characteristics of the included studies.

Study	Population	Country and Date	Follow-Up (Weeks)	No. of Patients
JANI et al. [17]	Patients with T2DM and hypertriglyceridemia are not on anti-dyslipidemia drugs, except for 10 mg of atorvastatin.	India, 2014	12	Saroglitazar 2 mg (*n* = 100) Saroglitazar 4 mg (*n* = 99) Placebo (*n* = 102)
PAI et al. [21]	Patients with T2DM with hypertriglyceridemia receiving sulphonylurea, metformin, or both for at least three months.	India, 2014	26	Saroglitazar 2 mg (*n* = 37) Saroglitazar 4 mg (*n* = 39) Pioglitazone 45 mg (*n* = 33)
GHOSH et al. [9]	Patients with diabetic dyslipidemia receiving 1000 mg of metformin daily.	India, 2015	12	Saroglitazar 4 mg (*n* = 18) Fenofibrate 160 mg (*n* = 18)
JAIN et al. [18]	Patients with T2DM with hypertriglyceridemia.	India, 2019	16	Saroglitazar 4 mg (*n* = 15) Placebo (*n* = 15)
KRISHNAPPA et al. [11]	Patients with T2DM with on a stable dose of metformin for at least six weeks.	India, 2020	12	Saroglitazar 2 mg (*n* = 380) Saroglitazar 4 mg (*n* = 386) Pioglitazone 30 mg (*n* = 389)
RASTOGI et al. [20]	Patients with T2DM and dyslipidemia on a stable dose of metformin.	India, 2020	12	Saroglitazar 4 mg (*n* = 15) Placebo (*n* = 15)
GAWRIEH et al. [12]	Patients with NAFLD not taking other lipid-lowering agents.	USA, 2021	16	Saroglitazar 2 mg (*n* = 25) Saroglitazar 4 mg (*n* = 27) Placebo (*n* = 28)
SIDDIQUI et al. [19]	Patients with NAFLD with and without statin therapy.	USA, 2023	52	Saroglitazar 4 mg (*n* = 130) Placebo (*n* = 91)

NAFLD: Nonalchoholic Fatty Liver Disease; T2DM: Type 2 diabetes mellitus.

### 3.2. Risk of Bias

Random sequence generation, allocation concealment, performance, and detection bias had minimal bias risks in 8/8 studies (100%). Decision bias (i.e., outcome assessment blinding) had a low risk in 6/8 studies (75%), but prejudice in reporting was identified in 4/8 (50%) studies. The “other bias” area looked at the financial sources, particularly from pharmaceutical industries, authors from pharmaceutical organizations, and conflicts of interest. Consequently, 4/8 studies (50%) had a risk of “other bias.” Figure 2 summarizes the risk of biased assessments.

### 3.3. Pooled Analysis of All Studies

Based on a thorough endpoint analysis, our primary efficacy outcomes were the triglyceride, LDL-C, and HDL-C levels. The secondary efficacy endpoints were total cholesterol, FPG, and Hb1Ac levels. The primary safety outcomes were creatinine and ALT levels and body weight (Table 2).

### 3.4. Efficacy Endpoints

#### 3.4.1. Triglycerides

A pooled analysis of six studies compared the triglyceride levels between the 4 mg saroglitazar (*n* = 418) and placebo (*n* = 372) groups [9,11,18,19,20,21]. The MD was −47.38 mg/dL (95% CI: −79.12 to −15.64 mg/dL; *p* = 0.03), demonstrating that 4 mg/day of saroglitazar decreased the triglyceride level compared to that in the control group in patients with dyslipidemia (Figure 3A).

We also compared the triglyceride level between the 2 mg (*n* = 256) and 4 mg (*n* = 270) saroglitazar groups. The MD was −32.38 mg/dL (95% CI: −53.62 to −11.14 mg/dL; *p* = 0.003), demonstrating that 4 mg/day of saroglitazar significantly decreased the triglyceride level compared to 2 mg/day in patients with diabetes-related dyslipidemia (Appendix A).

#### 3.4.2. HDL-C

A pooled analysis of five studies [11,17,18,20,21] compared the HDL-C level between the 4 mg saroglitazar (*n* = 357) and control (*n* = 356) groups. The MD was −1.38 mg/dL (95% Cl: −4.74 to 1.98 mg/dL; *p* = 0.42). The I^2^ value (93%) indicated a high level of heterogeneity among the studies (Figure 3B), suggesting that 4 mg/day of saroglitazar did not affect the HDL-C level compared to that in the control group in patients with dyslipidemia. The analysis comparing the 2 mg (*n* = 340) and 4 mg (*n* = 358) saroglitazar groups had similar results (MD: 0.03; 95% Cl: −2.29 to 2.35; *p* = 0.98; Appendix A).

#### 3.4.3. LDL-C

The pooled analysis of six studies [11,17,18,19,20,21] compared the LDL-C level between the 4 mg saroglitazar (*n* = 486) and control (*n* = 446) groups. The MD was −8.27 mg/dL (95% CI: −10.19 to −6.34 mg/dL, *p* < 0.00001; I^2^ = 73%; Figure 3C), demonstrating that 4 mg/day of saroglitazar significantly decreased the LDL-C level compared to that in the control group in patients with dyslipidemia.

Furthermore, a pooled analysis of 4 studies [11,12,17,21] compared the LDL-C level between the 2 mg (*n* = 340) and 4 mg (*n* = 358) saroglitazar groups; the LDL-C level did not differ between them (MD: −6.81 mg/dL; 95% CI: −14.53 to 0.90 mg/dL; *p* = 0.08; Appendix A).

#### 3.4.4. Total Cholesterol

A pooled analysis of six studies [11,17,18,19,20,21] compared the total cholesterol level between the 4 mg saroglitazar (*n* = 357) and control (*n* = 356) groups. The MD was −13.68 (95% CI: −16.69 to −10.66 mg/dL; *p* < 0.00001; I^2^ = 86%; Figure 3D), demonstrating that 4 mg/day of saroglitazar significantly decreased the total cholesterol level compared to that in the control group in patients with dyslipidemia.

In contrast, the total cholesterol level did not differ between the 2 mg (*n* = 315) and 4 mg (*n* = 331) groups (MD: −3.54 mg/dL; 95% CI: −10.60 to 3.53 mg/dL; *p* = 0.33; Appendix A).

#### 3.4.5. Apolipoprotein B

The pooled analysis of three studies [11,17,18] compared the apolipoprotein B level between the 4 mg saroglitazar (*n* = 307) and control (*n* = 315) groups. The MD was −4.14 (95% CI: −11.32 to 3.05; *p* = 0.26; Figure 3E), demonstrating that a 4 mg dose of saroglitazar did not affect the apolipoprotein B level compared to that in the control group in patients with dyslipidemia.

#### 3.4.6. Glucose Parameters

A pooled analysis of four studies [9,17,18,20] compared the FPG level between the 4 mg saroglitazar (*n* = 129) and control (*n* = 136) groups. The FPG level was significantly lower in the 4 mg saroglitazar group than in the control group (MD: −23.07; 95% CI: −32.07 to −14.08; *p* < 0.00001; Figure 3G). However, the HbA1c level did not differ between the groups (MD: −0.61 mg/dL; 95% CI: −1.47 to 0.25; *p* = 0.16; Figure 3F).

### 3.5. Safety Endpoints

#### 3.5.1. Serum Creatinine

A pooled analysis of three studies [12,17,21] compared the serum creatine levels between the 4 mg saroglitazar (*n* = 150) and control (*n* = 152) groups. The serum creatinine level significantly increased in the 4 mg saroglitazar group compared to that in the control group (MD: 0.12 mg/dL; 95% CI: 0.04 to 0.21 mg/dL; *p* = 0.004; Figure 4A). The serum creatinine level did not differ between the 2 mg (*n* = 123) and 4 mg (*n* = 125) saroglitazar groups (MD: −0.06; 95% CI: −0.15 to 0.04 mg/dL; *p* = 0.26; Appendix A).

#### 3.5.2. ALT

A pooled analysis of three studies [12,17,21] compared the ALT level between the 4 mg saroglitazar (*n* = 152) and control (*n* = 163) groups. The ALT level significantly decreased in the 4 mg saroglitazar group compared to that in the control group (SMD: −2.55; 95% CI: −4.62 to −0.48; *p* = 0.02; Figure 4B). However, the ALT level did not differ between the 2 mg (*n* = 150) and 4 mg (*n* = 150) saroglitazar groups (SMD: 0.91; 95% CI: −0.50 to 2.31; *p* = 0.21; I^2^ = 96%; Appendix A).

#### 3.5.3. Body Weight

A pooled analysis of four studies [11,17,18,21] compared the body weight of participants in the 4 mg saroglitazar (*n* = 330) and control (*n* = 354) groups; body weight did not differ between the groups (MD: −0.13 kg; 95% CI: −1.05 to 0.78; *p* = 0.77; I^2^ = 75%; Figure 4C). Body weight also did not differ between the 2 mg (*n* = 331) and 4 mg (*n* = 315) groups (MD: 0.35; 95% CI: −0.46 to 1.17; *p* = 0.40; I^2^ = 65%; Appendix A).

## 4. Discussion

This meta-analysis provides a contemporary, comprehensive review of the efficacy and safety of saroglitazar in individuals with dyslipidemia. We found that 4 mg of saroglitazar significantly decreased the triglyceride and LDL-C levels compared to those in the controls without increasing the HDL-C level. We also found that 4 mg of saroglitazar significantly increased the serum creatinine level, significantly decreased the ALT level, and did not affect body weight compared to that in the controls. 

Fibrates do not affect the HbA1c level (i.e., glycemia), whereas statins mildly increase it [19]. In this study, the FPG level was lower in the 4 mg saroglitazar group than in the anti-lipid, anti-diabetes, and placebo groups, reflecting the mild antihyperglycemic properties of saroglitazar.

Peroxisome proliferator-activated receptors are crucial in maintaining homeostasis, controlling inflammation, directing cell growth and differentiation, and limiting cell proliferation. [12] Despite its use as a lipid-lowering therapy, the PPAR-agonistic activity of fibrates is not very selective [19]. Fibrates cause serious illnesses, including myopathy, impaired renal function, and increased transaminase levels. Regarding PPAR-agonist action, thiazolidinediones reduce insulin resistance and blood glucose levels better than fibrates but can cause swelling and weight gain. [9].

In vitro and in vivo studies have investigated the pharmacodynamic action of saroglitazar in several disorders [22,23]. Our results suggest that 4 mg of saroglitazar may lower fat and glucose levels in patients with dyslipidemia, as evidenced by statistically significant reductions in total cholesterol and LDL-C levels without increasing the HDL-C level. These modifications are mostly attributable to the agonistic actions of PPAR, which are known to increase hepatic fatty acid oxidation and improve the lipid profile [19]. Increased HDL-C levels also suggest that saroglitazar might also be involved in reverse cholesterol metabolism, preventing the transfer of LDL-C. Our lipid profile results are consistent with those from studies of other glitazar-class medicines [24,25]. For instance, Dutta et al. found no change in the HbA1c or FBG levels after a comprehensive review and meta-analysis and reported that the effect of saroglitazar on glucose reduction varied based on the HbA1c level [10].

An observational study conducted by Shetty et al. [26] reported that adding saroglitazar to oral antidiabetic drugs resulted in a statistically significant improvement in glycemic (i.e., HbA1c) and lipid indices (i.e., triglycerides, total cholesterol, LDL-C, HDL-C, non-HDL, and very low-density lipoprotein cholesterol). The PPARα-agonist in this drug is responsible for the substantial triglyceride level decrease, while the PPARϒ-agonist is responsible for the HbA1c level decrease [23]. Our efficacy endpoint results indicated that saroglitazar significantly decreased fasting blood glucose, postprandial plasma glucose, and HbA1c levels, which is highly expressive of its ability to reduce endothelial aggression, a key factor in the onset of the process that ultimately leads to coronary disease [1]. Goyal et al. also reported that saroglitazar significantly reduced fasting blood glucose, postprandial plasma glucose, and HbA1c levels [27]. Furthermore, they also found decreased ALT, aspartate transaminase, alkaline phosphatase, and gamma-glutamyltransferase concentrations. However, only alkaline phosphatase and gamma-glutamyltransferase levels significantly decreased among individuals with cardiometabolic illnesses treated with saroglitazar [12]. Previous research has shown that the ALT level in patients treated with saroglitazar consistently improved [20,28].

The recent PROMINENT trial demonstrated that the incidence of cardiovascular events remained comparable between patients who received fibrates and those given a placebo. Although levels of triglycerides, VLDL cholesterol, remnant cholesterol, and apolipoprotein C were lower in the fibrate group, there was no reduction in apolipoprotein B levels, and it was felt that this medication does not improve cardiovascular outcomes [7]. In light of this, we emphasize that in our analysis, saroglitazar did not significantly reduce apolipoprotein B, highlighting that in spite of the fact that there were improvements in the saroglitazar group regarding the lipid profile, this may not translate to improved cardiovascular outcomes.

Saroglitazar is not excreted via the kidneys, and thus, some adverse effects have been observed in clinical studies, including hepatotoxicity and renal toxicity [12,26]. We identified a significant association between 4 mg of saroglitazar and increased serum creatinine levels (MD: 0.12 mg/dL, *p* = 0.004). However, we cannot find evidence to explain this result. One RCT assessed the estimated glomerular filtration rate (GFR), which did not decrease. Thus, consistent with the study by Dutta et al., our meta-analysis results raise concerns regarding saroglitazar and renal safety [10]. Tesaglitazar and aleglitazar increase serum creatinine, blood urea nitrogen levels, and the GFR [29]. Thus, long-term saroglitazar users should be monitored for changes in the serum creatinine level, uremic indices, urine microalbumin, and renal architecture for at least one year. Renal hemodynamics may cause renal tubules to synthesize and secrete more creatinine [12]. However, cystatin-C, inulin clearance, and GFR markers are unaffected by fenofibrate [9]. Saroglitazar, which has a PPAR-alpha agonist action similar to fenofibrate, may have comparable effects on serum creatinine levels.

Krishnappa et al. [11] examined blood creatinine levels over 56 weeks and found no significant differences between saroglitazar and pioglitazone users. Our data suggest that saroglitazar patients require a specialized and well-powered study that examines renal parameters, including glomerular filtration rate and urine microalbumin/creatinine ratio. Finally, long-term adverse effects such as cardiovascular disease (which requires a dedicated cardiovascular outcome trial), bladder cancer (seen with some other peroxisome proliferator-activated receptor (PPAR) agonists), and liposarcomas (seen with muraglitazar) must be fully evaluated to ensure the long-term safety of saroglitazar in clinical practice [27,30,31,32,33].

With regard to future perspectives, the pleiotropic effects of saroglitazar, which affect both glucose and lipid metabolism, play an important role in its potential as a therapeutic option. Sasso et al. [34] examined type 2 diabetes subjects with albuminuria and a history of cardiovascular disease and reported that the increase in the number of risk factors at target correlates with better cardiovascular outcomes in patients with type 2 DM at high CV risk. That being so, saroglitazar could represent a potential new therapeutic option. 

This study has some limitations. First, all the included studies were clinical trials, which often have a small sample size and a short follow-up time, making it difficult to determine the long-term safety and effectiveness of saroglitazar. Second, since all the included trials were performed in India, these results may not be generalizable to all individuals with dyslipidemia. Third, heterogeneity is greater than 50% in many of the outcome analyses, even though the number of studies is limited to fewer than ten (Appendix A). Finally, every experiment that was considered had a shorter follow-up period with surrogate results.

## 5. Conclusions

This meta-analysis suggests that 4 mg of saroglitazar reduces triglyceride, LDL-C, and total cholesterol levels in patients with dyslipidemia; however, there were no significant changes in apolipoprotein B levels. Thus, it improves the lipid profile; nevertheless, it may not reduce cardiovascular outcomes. Furthermore, it significantly increased the creatinine level, which is a potential safety concern.

## Figures and Tables

**Figure 1 jcm-12-05674-f001:**
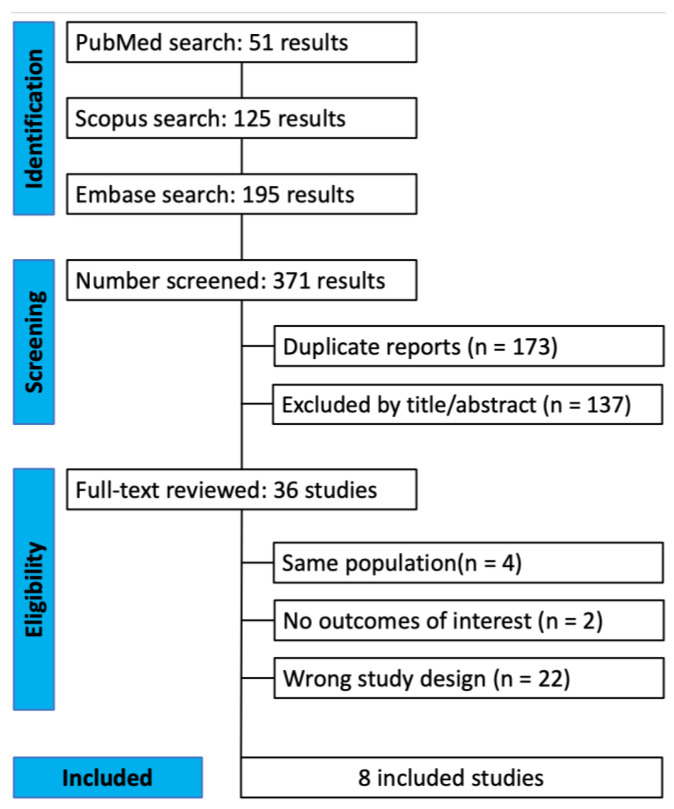
Preferred Reporting Items for Systematic Reviews and Meta-Analyses: Flow Diagram of the Study Screening and Selection Process.

**Figure 2 jcm-12-05674-f002:**
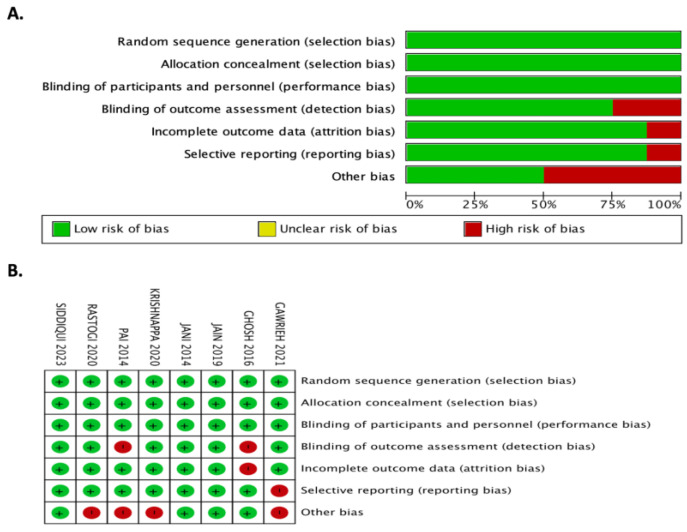
Risk of bias assessments (**A**) Graphical representation of the risk of bias (percentages) (**B**) Summary of the risk of bias. Risk of Bias (RoB) plot. Circles are colour coded with green indicating low RoB, yellow indicating some concerns and red indicating a high RoB. The symbols “+”, “?”and “-“ indicate the same RoB grades as the colours.

**Figure 3 jcm-12-05674-f003:**
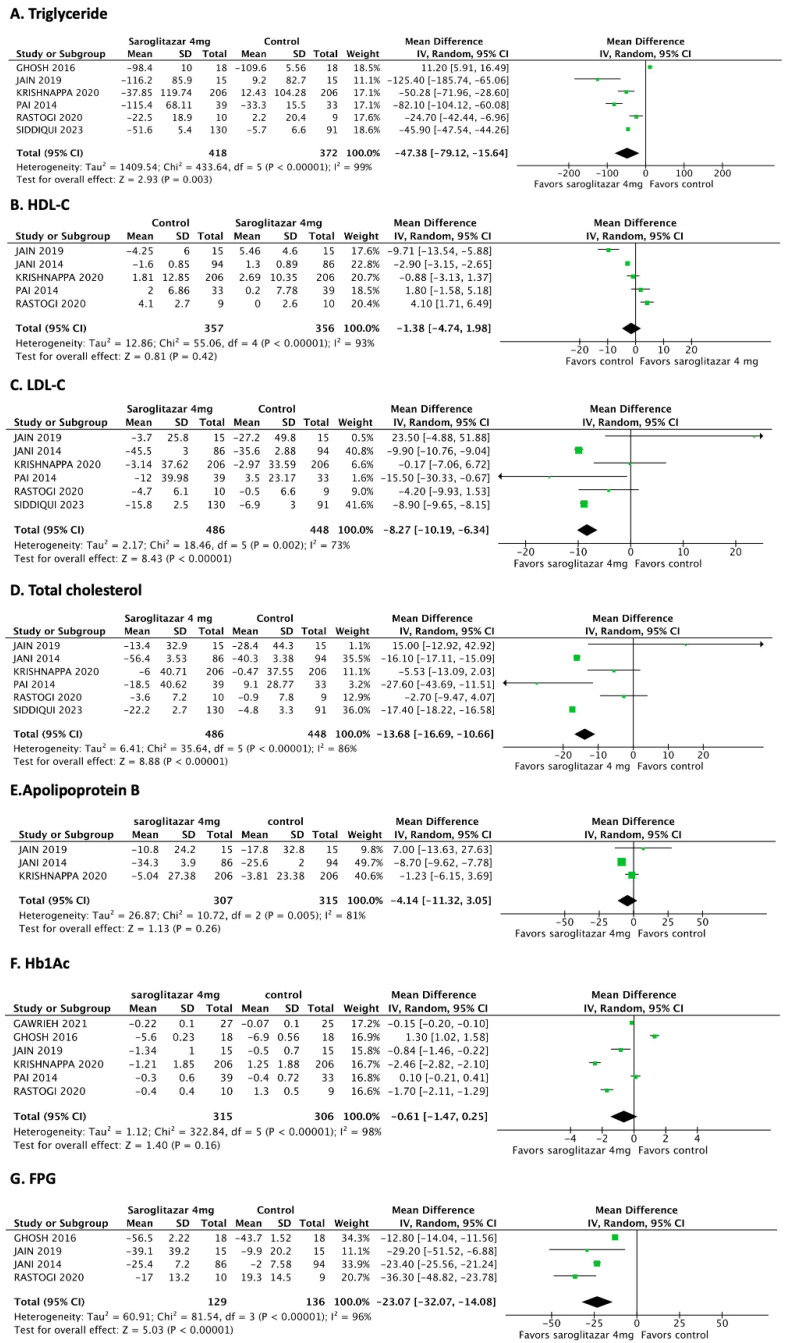
Forest plots comparing the efficacy endpoints between the 4 mg saroglitazar and control groups Saroglitazar (**A**) significantly reduces the triglyceride level [9,11,18,19,20,21]; (**B**) does not affect the high-density lipoprotein cholesterol level [11,17,18,20,21]; (**C**) significantly reduces the low-density lipoprotein and [11,17,18,19,20,21] (**D**) total cholesterol levels [11,17,18,19,20,21]; (**E**) does not affect the apolipopotein B levels [11,17,18]; (**F**) does not affect the hemoglobin A1c level [9,11,12,18,20,21]; and (**G**) significantly decreases the fasting plasma glucose level compared to the control [9,17,18,20]. CI: confidence interval; IV: inverse variance; SD: standard deviation; The green squares represent the weighted mean difference (WMD) of each study, the horizontal line represents 95% confidence intervals (95% CI), and the black diamond represents the summary of weight mean difference.

**Figure 4 jcm-12-05674-f004:**
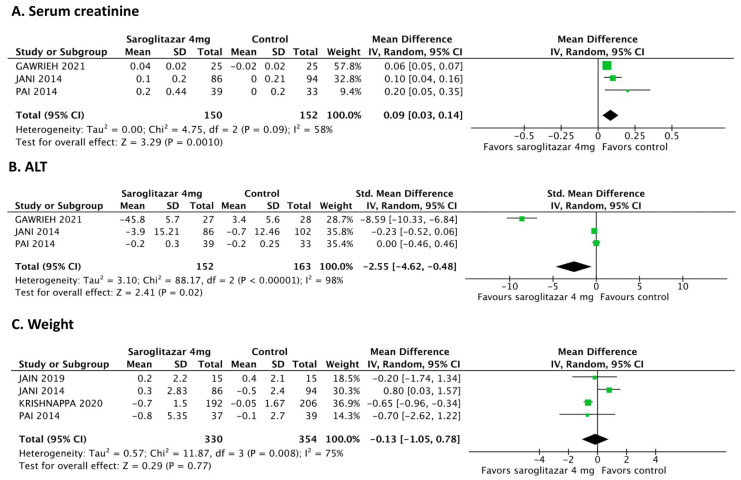
Forest plots comparing the safety endpoints between the 4 mg saroglitazar and control groups Saroglitazar (**A**) significantly increased the serum creatine level [12,17,21]; (**B**) significantly decreased the alanine transaminase level [12,17,21]; and (**C**) did not affect body weight compared to the control group [11,17,18,21]. CI: confidence interval; IV: inverse variance; SD: standard deviation. The green squares represent the weighted mean difference (WMD) of each study, the horizontal line represents 95% confidence intervals (95% CI), and the black diamond represents the summary of weight mean difference.

**Table 2 jcm-12-05674-t002:** Baseline characteristics of patients in included studies.

Study	Groups	Demographic Profile		Clinical Profile		Laboratory Parameters	Lipid Parameters (mg/dL)				Safety Parameters	
		Age	Sex (F/M)	Weight (kg)	Height (cm)	HbA1c (%)	TG	TC	HDL-C	LDL-C	Cr (mg/dL)	ALT (U/L)
JANI et al. [17]	Saroglitazar 2 mg (*n* = 100)	50.4 ± 9.01	39/61	71.3 ± 13.56	161.9 ± 9.66	8.9 ± 1.84	273.3 ± 78.58	200.6 ± 38.11	36.6 ± 8.45	132.5 ± 30.43	NR	NR
	Saroglitazar 4 mg (*n* = 99)	51.2 ± 8.66	43/56	69.1 ± 10.83	160.5 ± 9.06	8.9 ± 1.77	287.3 ± 85.94	210.4 ± 37.20	39.1 ± 11.19	140.2 ± 29.36	NR	NR
	Placebo (*n* = 102)	49.8 ± 9.95	47/55	69.9 ± 11.53	160.9 ± 8.66	9.2 ± 1.81	286.6 ± 78.92	209.5 ± 39.31	38.5 ± 12.06	140.1 ± 33.58	NR	NR
PAI et al. [21]	Saroglitazar 2 mg (*n* = 41)	48.9 ± 8.98	15/26	69.8 ± 12.72	161.9 ± 9.44	8.1 ± 0.86	253.9 ± 68.44	202.4 ± 47.60	36.8 ± 12.09	134.8 ± 42.56	0.7 ± 0.21	31.5 ± 16.48
	Saroglitazar 4 mg (*n* = 41)	47.3 ± 9.10	16/24	73.0 ± 11.49	163.1 ± 10.17	7.9 ± 0.58	257.0 ± 52.39	197.3 ± 40.98	35.3 ± 9.64	130.8 ± 38.83	0.7 ± 0.19	29.7 ± 15.91
	Pioglitazone 45 mg (*n* = 40)	49.9 ± 10.98	16/24	71.0 ± 12.94	162.0 ± 10.74	8.2 ± 0.75	265.0 ± 61.66	185.8 ± 29.91	38.3 ± 10.85	116.6 ± 29.25	0.7 ± 0.2	26.3 ± 9.13
GHOSH et al. [9]	Fenofibrate 160 mg (*n* = 18)	58.1	20/25	NR	NR	7.1 ± 0.4	244.2 ± 20.6	NR	42.12 ± 5.19	114.1 ± 7.11	NR	NR
	Saroglitazar 4 mg (*n* = 18)	62.6	NR	NR	NR	6.9 ± 0.6	245.9 ± 33.9	NR	40.18 ± 5.89	114.2 ± 10.76	NR	NR
JAIN et al. [18]	Saroglitazar 4 mg (*n* = 15)	40.9 ± 9.6	0/15	78.7 ± 9.8	169.7 ± 5.6	NR	325.6 ± 129.3	192.4 ± 42.9	37.49 ± 9.6	116.4 ± 36.3	NR	NR
	Placebo (*n* = 15)	47 ± 8.8	3/12	75.6 ± 11.0	164.5 ± 11.2	NR	236.3 ± 83.1	217.6 ± 45.4	45.3 ± 8.5	146.7 ± 45.3	NR	NR
KRISHNAPPA et al. [11]	Saroglitazar 2 mg (*n* = 380)	51.90 ± 10.38	164/216	70.27 ± 11.84	NR	9.76 ± 1.59	163.87 ± 91.49	176.98 ± 42.67	42.39 ± 10.58	117.11 ± 36.92	NR	NR
	Saroglitazar 4 mg (*n* = 386)	51.34 ± 10.06	143/243	69.09 ± 11.46	NR	9.72 ± 1.58	172.52 ± 123.67	174.03 ± 39.32	41.50 ± 10.47	112.93 ± 34.89	NR	NR
	Pioglitazone 30 mg (*n* = 389)	51.84 ± 9.76	167/222	69.49 ± 11.59	NR	9.49 ± 1.54	166.20 ± 89.93	176.42 ± 37.83	42.64 ± 12.72	116.77 ± 32.31	NR	NR
RASTOGI et al. [20]	Saroglitazar 4 mg (*n* = 15)	53.1 ± 8.8	8/7	69.9 ± 12.6	159.3 ± 10.3	7.6 ± 0.9	NR	176.7 ± 41.4	37.7 ± 7.6	117.4 ± 38.4	NR	NR
	Placebo (*n* = 15)	54.9 ± 7.8	6/9	78.0 ± 11.7	164.1 ± 9.9	8.0 ± 1.0	NR	151.4 ± 36.4	47.4 ± 8.8	89.0 ± 36.3	NR	NR
GAWRIEH et al. [12]	Saroglitazar 2 mg (*n* = 25)	47.9 ± 10.4	12/13	NR	NR	6.8 ± 1.5	201.9 ± 116.6	194.0 ± 44.0	44.5 ± 7.3	124.3 ± 36.9	0.8 ± 0.2	84.8 ± 29.3
	Saroglitazar 4 mg (*n* = 27)	49.0 ± 11.0	12/15	NR	NR	6.1 ± 0.9	190.9 ± 98.5	204.8 ± 62.3	46.8 ± 15.8	132.7 ± 56.1	0.8 ± 0.2	83.4 ± 27.9
	Placebo (*n* = 28)	48.7 ± 10.5	13/15	NR	NR	6.2 ± 1.0	181.1 ± 62.2	191.7 ± 39.7	46.9 ± 12.0	121.6 ± 38.1	0.8 ± 0.2	93.4 ± 42.1
SIDDIQUI et al. [19]	Saroglitazar 4 mg (*n* = 130)	48.0 ± 0.9	53/77	NR	NR	6.2 ± 1.0	182.4 ± 116.1	188.8	43.3	117.5	NR	76 ± 51
	Placebo (*n* = 91)	47.8 ± 10.1	44/47	NR	NR	6.0 ± 0.9	171.6 ± 68.2	185.3	43.8	115.4	NR	72 ± 42

ALT: alanine transaminase; Cr: Creatine; NR: Not reported; TC: Total cholesterol; TG: Triglycerides.

## Data Availability

The data underlying this article will be shared on reasonable request with the corresponding author.

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
