# Peer review of "Dual PPRαϒ Agonists for the Management of Dyslipidemia: A Systematic Review and Meta-Analysis of Randomized Clinical Trials"

_jcm, 2023, doi:10.3390/jcm12175674_

Round 1
Reviewer 1 Report
I read with great interest the article titled "Dual PPRαϒ agonists for the management atherogenic dyslipidemia: A metanalysis of randomized clinical trials" by Antonio da Silva Menezes Junior et al.
The paper's design is sound, and the article is logically organized into appropriate sections and subsections. English is generally fine, only minor spell check needed.
Here are the comments and suggested revisions:
1. Title: as per the method section, the paper is categorized as a systematic review and meta-analysis. Could you please adjust the relevant information accordingly?
2. Introduction: It appears that there may be a misunderstanding from title to the introduction, as the focus of the paper does not revolve around diabetic dyslipidemia. Even though 6 out of 8 studies included in the analysis are performed on diabetic subjects and the drug is approved for the management of diabetic hypertriglyceridemia in India. To provide clarity, I kindly request a modification of the introduction and/or of the title.
3. Future perspective: The pleiotropic effect of saroglitazar affecting both glucose and lipid metabolism could represent a potential new therapeutic option (even though still not commercially available in many countries), as always new evidence suggests the role of multifactorial interventions for the amelioration of clinical outcomes, especially in diabetic patients (doi: 10.1186/s12933-022-01674-7). Please report it in the discution section, in order to further enhance your findings.
Revision is required
Author Response
RESPONSE TO REVISOR #1
I read with great interest the article titled "Dual PPRαϒ agonists for the management atherogenic dyslipidemia: A metanalysis of randomized clinical trials" by Antonio da Silva Menezes Junior et al.
The paper's design is sound, and the article is logically organized into appropriate sections and subsections. English is generally fine, only minor spell check needed.
We express our satisfaction in having our manuscript meticulously reviewed and extend our gratitude to the reviewer for diligently acknowledging the methodological rigor employed in the composition of this paper. The work has undergone professional editing by Editage to ensure accuracy in language and grammar. The certificate of editing is provided on the website.
Here are the comments and suggested revisions:
- Title: as per the method section, the paper is categorized as a systematic review and meta-analysis. Could you please adjust the relevant information accordingly?
Thank you for providing your suggestion. The paper title was modified to align with its designated category, namely systematic review and meta-analysis. The current title of the manuscript is "Dual PPRαϒ agonists for the treatment of dyslipidemia: a systematic review and meta-analysis of randomized clinical trials."
- Introduction: It appears that there may be a misunderstanding from title to the introduction, as the focus of the paper does not revolve around diabetic dyslipidemia. Even though 6 out of 8 studies included in the analysis are performed on diabetic subjects and the drug is approved for the management of diabetic hypertriglyceridemia in India. To provide clarity, I kindly request a modification of the introduction and/or of the title.
We empathize with your proposition. The paper title and introduction section have been modified in accordance with your suggestion, shifting the emphasis away from "diabetic dyslipidemia" as it is not the central theme of the entire paper. The alterations related to this subject were marked with yellow highlighting. In order to enhance the quality of our article in this regard, modifications were made to the manuscript title, introduction, and methods section, particularly in relation to the section titled "2.1 Study Eligibility Criteria." 2.1. Study eligibility criteria
Our primary study question was: “What is the safety and effectiveness of saroglitazar for treating dyslipidemia?” Our investigation was conducted based on the following population, intervention, comparison, outcome, and time (study design) (PICOT) categories: Population (i.e., P): adult patients with diabetic hypercholesterolemia or dyslipidemia;
- Future perspective: The pleiotropic effect of saroglitazar affecting both glucose and lipid metabolism could represent a potential new therapeutic option (even though still not commercially available in many countries), as always new evidence suggests the role of multifactorial interventions for the amelioration of clinical outcomes, especially in diabetic patients (doi: 10.1186/s12933-022-01674-7). Please report it in the discussion section, in order to further enhance your findings.
We acknowledge the significance of this observation. We concur with the reviewer's assertion that this point holds significant importance. During our conversation, we incorporated an examination of the multifactorial therapies' function in improving clinical outcomes. We specifically emphasized the pleiotropic effect of saroglitazar and its significance in shaping future perspectives Discussion
With regards to future perspectives, the pleiotropic effects of saroglitazar affecting both glucose and lipid metabolism plays an important role on its potential as a therapeutic option. Sasso et al. [34] examined type 2 diabetes subjects with albuminuria and history of cardiovascular disease and reported that the increase in the number of risk factors at target correlates with better cardiovascular outcomes in patients with type 2 DM at high CV risk. That being so, saroglitazar could represent a potential new therapeutic option.

Reviewer 2 Report
The current study is a meta-analysis of saroglitazar with regard to its lipid lowering efficacy and safety. The authors have performed a well-done meta-analysis. Their meta-analysis showed that saroglitazar can significantly lower triglycerides though there are some safety concerns. However, what is most important is will this benefit in lipid lowering translate to improved cardiovascular outcomes? There have been several studies now evaluating fibrate therapy and cardiovascular outcomes and none have shown that fibrate therapy, although lowering triglycerides improves cardiovascular outcomes. It is important both in the introduction and in the discussion of this manuscript that these points be mentioned.
The recent PROMINENT study showed that pemafibrate lowers triglycerides in high risk diabetic patients but did not improve cardiovascular outcomes. In the PROMINENT study although triglycerides were lowered, apolipoprotein B was not and it was felt that the reason that this medication did not improve cardiovascular outcomes is that the number of atherogenic apolipoprotein B containing particles were not significantly reduced. The authors need to discuss the PROMINENT trial in their discussion. Not all drugs that improve the dyslipidemia associated with triglycerides translates into a reduction in cardiovascular events.
Specific comments
1. Did any of the studies included in the meta-analysis measure apolipoprotein B?. If so was this reduced with saroglitizar? If so this should be mentioned in the manuscript and if not this should be discussed in the limitations section of the manuscript.
Author Response
RESPONSE TO REVISOR #2
The current study is a meta-analysis of saroglitazar with regard to its lipid lowering efficacy and safety. The authors have performed a well-done meta-analysis. Their meta-analysis showed that saroglitazar can significantly lower triglycerides though there are some safety concerns. However, what is most important is will this benefit in lipid lowering translate to improved cardiovascular outcomes? There have been several studies now evaluating fibrate therapy and cardiovascular outcomes and none have shown that fibrate therapy, although lowering triglycerides improves cardiovascular outcomes. It is important both in the introduction and in the discussion of this manuscript that these points be mentioned.
Thanks a lot. I has been done.
The recent PROMINENT study showed that pemafibrate lowers triglycerides in high risk diabetic patients but did not improve cardiovascular outcomes. In the PROMINENT study although triglycerides were lowered, apolipoprotein B was not and it was felt that the reason that this medication did not improve cardiovascular outcomes is that the number of atherogenic apolipoprotein B containing particles were not significantly reduced. The authors need to discuss the PROMINENT trial in their discussion. Not all drugs that improve the dyslipidemia associated with triglycerides translates into a reduction in cardiovascular events.
We express our gratitude for the meticulous examination that you and the reviewers dedicated to our article. We express our gratitude for the opportunity to have this amended version of our work considered for publication in your prestigious journal. In the subsequent discussion, we have provided our explicit solutions to the inquiries made in your comprehensive analysis. Furthermore, we have indicated the modifications made to the document by marking them in yellow, in accordance with the suggestions provided by you. The revisions made to our manuscript are believed to have enhanced its overall quality. We sincerely hope that this revised version meets the standards required for publication and is deemed acceptable by the reviewers.
Specific comments
- 1. Did any of the studies included in the meta-analysis measure apolipoprotein B?. If so was this reduced with saroglitizar? If so this should be mentioned in the manuscript and if not this should be discussed in the limitations section of the manuscript.
I express my gratitude for your valuable contribution. We concur with the reviewer's assertion that this issue holds significant importance, particularly in regards to the correlation between apolipoprotein B and the mitigation of cardiovascular consequences. In the introduction and discussion sections, we included the observation that the PROMINENT research demonstrated a reduction in triglyceride levels, but did not yield significant cardiovascular benefits with the use of pemafibrate. The lack of reduction in apolipoprotein B was identified as a significant factor contributing to the absence of reduced cardiovascular outcomes.
Furthermore, under the results section (3.4.5 Apolipoprotein B), we incorporated a combined analysis of three research pertaining to apolipoprotein B, and no statistically significant alterations were identified. The aforementioned findings were also deliberated over in the discussion and conclusion portion of our study, whereby we emphasized the possibility that the observed decrease in triglyceride levels within the saroglitazar group may not necessarily correspond to a reduction in cardiovascular events.
Results
3.4.5. Apolipoprotein B
The pooled analysis of 3 studies [11,17,18] compared the apolipoprotein B level between the 4mg saroglitazar (n= 307) and control (n= 315) groups. The MD was -4.14 (95% CI; -11.32 to 3.05; p=0.26; Figure 3E), demonstrating that a 4mg dose of saroglitazar did not affect the apolipoprotein B level compared to that in the control group in patients with dyslipidemia.
Discussion
The recent PROMINENT trial demonstrated that the incidence of cardiovascular events remained comparable between patients who received fibrates and those given a placebo. Although levels of tryglicerides, VLDL cholesterol, remnant cholesterol, and apolipoprotein C were lower in the fibrate group, there was no reduction in apolipoprotein B levels, and it was felt as the reason for this medication do not improve cardiovascular outcomes [7]. In the light of this, we emphasize that in our analysis, saroglitazar did not significantly reduced apolipoprotein B, highlighting that in spite of the fact that there were improvement in the saroglitazar group regarding the lipid profile, this may not translate into improved cardiovascular outcomes.
Conclusion
This meta-analysis suggests that 4 mg of saroglitazar reduces the triglyceride, LDL-C, and total cholesterol levels in patients with dyslipidemia, however, there were no significant changes in apolipoprotein B levels, thus, it improves lipid profile, nevertheless, it may not reduce cardiovascular outcomes. Furthermore, it significantly increased the creatinine level, which is a potential safety concern.
We express our gratitude for providing us with the chance to enhance our text through your invaluable comments and inquiries. We have diligently endeavored to integrate your input and trust that these improvements will effectively convince you to accept our submission.